# OpenReview forum: "Practical alignment requires more than learning from human feedback"
_ICLR.cc/2025/Conference — Submitted to ICLR 2025_

### Official Review · Reviewer_6BJn · 2024-11-01

**Soundness:** 3
**Presentation:** 3
**Contribution:** 3
**Rating:** 8
**Confidence:** 4

**Summary:**

The authors introduce "practical alignment," an AI framework designed to address limitations in current alignment methods, especially Reinforcement Learning from Human Feedback (RLHF). RLHF assumes that human feedback accurately reflects true desires and remains stable, leading to potential misalignment when human beliefs are flawed or change over time. The authors argue that effective AI alignment should adapt to these variabilities and that AI should not only learn from but also educate humans to correct misunderstandings. They introduce the MindGrid toolkit, which allows simulation of practical alignment scenarios, including a teaching component where the AI informs humans about reality to adjust their beliefs. Experimental results on language models demonstrate the importance of teaching in achieving alignment and reveal current limitations in language models' teaching abilities.

**Strengths:**

-The practical alignment framework is an innovative approach that considers human belief variability, introducing a more realistic and dynamic model for AI-human alignment.
- The MindGrid toolkit is a valuable contribution, providing researchers with a means to simulate complex alignment scenarios that involve misunderstandings and belief updates.
-The paper advances AI alignment by emphasizing the importance of an AI’s teaching role, which could make AI systems more effective collaborators.
- The authors conduct extensive tests with different language models, showing the practical challenges and gaps in current alignment techniques.

**Weaknesses:**

- While addressing variability in beliefs, the model assumes a static "real world" that may not hold in all domains, such as those influenced by subjective or evolving truths.
- The experiments are limited to simulated environments, which might not fully capture the complexity of real-world human-AI interactions, particularly where human beliefs vary significantly.

**Questions:**

- How would practical alignment handle environments where the "real world" is itself subjective or dynamic, such as in social or cultural contexts?
- How does the model account for situations where human feedback may be influenced by biases or lack of knowledge rather than misunderstandings about objective facts?

---

> ### Author Response · Authors · 2024-11-22
> **Thank you for your feedback**
>
> We thank you for your appreciation of the paper. For your two questions, they are excellent for future exploration. We acknowledge the limitations of our framework in modeling such scenarios (lines 231–235) and have raised similar questions in the conclusion section. Addressing these questions may require insights from other fields, such as social sciences. Despite its current limitations, we believe our framework represents a significant improvement over existing ones and can inspire readers to think about important questions like these.

---

### Official Review · Reviewer_8woS · 2024-11-02

**Soundness:** 3
**Presentation:** 2
**Contribution:** 3
**Rating:** 5
**Confidence:** 4

**Summary:**

This paper provides conceptual frameworks for 'ostensible' and 'practical' alignment. It shows that techniques like RLHF promote ostensible alignment as they incentivize performing best with respect to the world model of the human principal. However, the practical alignment objective which optimizes human desire function in the real world environment is more robust.

**Strengths:**

The topic of the paper is very interesting and challenging. I think the authors did a good job at developing conceptual frameworks that are insightful. The main strength of the paper is that it has done an excellent job at formalizing lots of informal intutitions that people who think about and work on AI alignment have. If I did not have concerns around the presentation of the paper, I would have likely recommended a strong accept (my current recommendation is weak reject).

**Weaknesses:**

- The paper is hard to follow. This is my major issue imo. This paper is very interesting, but I am afraid that if the presentation is not accessible, it will fail to have much impact. See my notes on presentation below.
- The discussion of related works is quite shallow, and an important related work in my opinion is not covered (https://arxiv.org/abs/1711.02827).
- The paper does not propose a method for 'practical' alignment; this is not strictly a weakness but lack of a method does weaken the contributions the paper is making a little.

#### Partial notes on presentation (these also double as questions in some cases)
*Section 3.1*
- The paper directly dives into describing what the ostensible alignment is without presenting the setup being assumed. Is it an MDP, POMDP, CIRL like Dec-POMDP setup? My sense is it is CIRL like Dec-POMDP setup but I am not super confident in that.
- The paper is often introducing terminology without properly defining them. For example, second para of section 3.1 introduces the concept of plan without defining what the plan is.
- In description of ostensible alignment process, it is not clear to me how to think about 'conversation context'. Is it something like 'initial state' in MDP framework?
- Bold p is used for joint policy, but for individual policies $p^A$, $p^H$, the p used is not bold. Authors should be consistent in their use of bold. I would also suggest avoiding the use of p and using some greek/latin symbol for policy as p is sort of a universal symbol for policy distribution. The convention in RL is to use \pi, but the authors are using it already.
- Equation 1 was highly confusing for me, RHS is the definition of J^H(p) but it looks as the definition of max J^H(p)..
- \mathcal{U^H}, \mathcal{U^A} are never defined.. presumably they are the sets from which utterances are sampled?
- I think if the authors lead the discussion with some sort of example that grounds these concepts, that would make it much easier for the reader to understand them.
- Would not it make sense to start by introducing practical alignment; and then discussing 'ostensible' alignment as a simpler case? Seems like the main (only?) diff btw ostensible and practical alignment is whether the $\theta$ used in the reward function is $\theta^*$ or $\theta^H$.

*Section 3.2*
- Defining $(\psi^H, w^*)$ to be $theta^*$ does not seem like a great choice to me; it is probably not saving you much space so why not just have $\psi^H, w^*$ directly in the expressions?
- Definition 3.1 should be a lemma.

**Questions:**

1. Are the authors aware of inverse reward design (https://arxiv.org/abs/1711.02827)? It is probably the most related work to this paper that I can think of; but has not been cited in the paper. I would like authors to include discussion on novelty of this work relative to IRD. In general, the related works section is relatively thin for my liking with most citations after 2020. The 'teaching' aspect of the framework is also related to works on 'eliciting latent knowledge' (https://www.lesswrong.com/posts/qHCDysDnvhteW7kRd/arc-s-first-technical-report-eliciting-latent-knowledge), debate between agents (https://arxiv.org/abs/2402.06782) and eliciting human preferences better (https://arxiv.org/abs/2310.11589).

2. I don't think 'practical alignment' is a great name for the concept/framework being introduced in the paper. In fact, it seems to be harder to implement and thus seems a bit impractical. Authors should rename it to something like 'idealized alignment'.

3. In practical alignment, does the agent observe $\omega^*$? This is not explicitly stated but based on the description of the experiments, seems to be assumed. This seems like a very strong assumption and should be made explicit and acknowledged as such.

4. In equation 4 the meaning of $\theta^A$ is not obvious to me. Similarly, the meaning of subscript of T in $\theta_T^Z$ is also not clear.

5. Can you give a scaling curve for this? My guess is that performance would change abruptly past some threshold.
>Interestingly, adding training examples generally worsens model performance.

---

> ### Author Response · Authors · 2024-11-22
> **Thank you for your feedback**
>
> We appreciate your evaluation of the paper. While the presentation can be improved, we hope you consider that the problem we study is novel and may not immediately align with conventional thinking. Although readers may initially find the concepts challenging to grasp, the framework's impact could be enduring. In the next revision, we will include more examples to clarify the concepts and simplify the notation.
>
> Below, we address your major concerns and questions:
>
> > The paper does not propose a method for 'practical' alignment; this is not strictly a weakness but lack of a method does weaken the contributions the paper is making a little.
>
> We actually proposed a solution: that is to design AI systems that can detect human false beliefs and generate language utterances to correct those beliefs. Such systems differ dramatically from existing ones, which only passively learn from humans. Using LLMs to infer false beliefs and generate language utterances, we obtained higher results than a baseline that cannot influence humans (ostensible alignment). This paper is already dense so further improvement warrants an entire new paper.
>
> > 1. Are the authors aware of inverse reward design (https://arxiv.org/abs/1711.02827)? It is probably the most related work to this paper that I can think of; but has not been cited in the paper. I would like authors to include discussion on novelty of this work relative to IRD. In general, the related works section is relatively thin for my liking with most citations after 2020. The 'teaching' aspect of the framework is also related to works on 'eliciting latent knowledge' (https://www.lesswrong.com/posts/qHCDysDnvhteW7kRd/arc-s-first-technical-report-eliciting-latent-knowledge), debate between agents (https://arxiv.org/abs/2402.06782) and eliciting human preferences better (https://arxiv.org/abs/2310.11589).
>
> We thank you for your great suggestions. We did cite Li et al.'s work on eliciting human preferences. We will add more works to the related work section. Nevertheless, we emphasize that they are remotely related to our work because we focus on a theoretical formulation of the problem rather than empirical approaches.
>
> IRD highly related, thanks for pointing it out! The fundamental differences between our work and IRD are:
> - IRD assumes that **the AI knows the human’s world model, so its job is only to infer their desire function**. In their experiments, the AI is trained in the environment (world model) from which the human observes and derives their proxy reward. This is an unrealistic model of communication, in which the mental representations (belief, intention) of one agent is hidden from the other. Our framework captures this aspect of communication. In our experiments, the AI has to infer the world model of the human.
> - IRD does NOT model world-model change, while our framework does.
> - IRD’s solution focuses on uncertainty, while we propose a communication-based solution in which the AI actively informs the human about the world.
>
> > I don't think 'practical alignment' is a great name for the concept/framework being introduced in the paper. In fact, it seems to be harder to implement and thus seems a bit impractical. Authors should rename it to something like 'idealized alignment'.
>
> “Practical alignment” is actually a shorthand for “alignment for practical outcomes”, i.e. outcomes that succeed in the real world. It describes what is encoded in the objective of the framework. It contrasts with the existing alignment frameworks which may produce solutions that fail in the real world. We will think of a different term to avoid this confusion.
>
> >In practical alignment, does the agent observe ω∗? This is not explicitly stated but based on the description of the experiments, seems to be assumed. This seems like a very strong assumption and should be made explicit and acknowledged as such.
>
> The AI does observe omega^* in our experiments but not in a general practical alignment problem. It was clearly stated in lines 431-432.
>
> >In equation 4 the meaning of θ^A is not obvious to me. Similarly, the meaning of subscript of T in θ_T^Z is also not clear.
>
> We couldn’t find  θ^A in equation 4. We have defined θ_t^A and θ_t^H in lines 213-221. The meaning θ^T_Z can be inferred from those definitions but we will add a reminder.
>
> > Can you give a scaling curve for this? My guess is that performance would change abruptly past some threshold.
>
> Could you clarify the scale you are referring to (number of examples or model size)? In table 1, we have ordered the models roughly by sizes and presented results with different numbers of examples.

---

> > ### Comment · Reviewer_8woS · 2024-11-24
> >
> > The proposed method is not going to work in any practical scenarios; if author's think it would, they need to demonstrate it. Doing so would significantly strengthen the paper. Another reason for not thinking this method is practical is that you are assuming access to $\omega^*$ in your experiments, which is clearly very unrealistic.
> >
> > The scaling curve thing I want is for # of few shot examples.
> >
> > In general, I am disappointed that the authors' have not engaged with my critique around presentation issues in the paper that make it hard to follow. Ensuring that your work is well-presented and accessible is an important part of the publication process in my opinion. I agree that the problem you are studying is novel and is not well-aligned with the conventional thinking, but that makes it *more* important to ensure that the paper presentation is as best as it could be. Given there has not been any improvement on this front, I will be keeping my current score.

---

> > > ### Author Response · Authors · 2024-11-30
> > > **Thank you for response**
> > >
> > > We appreciate the time and effort you spent on evaluating our paper. We would like to reply to some of your points. Please understand that we are not pressuring to change your evaluation, but we think these clarifications might be helpful for the AC to make the final decision.
> > >
> > > > The proposed method is not going to work in any practical scenarios; if author's think it would, they need to demonstrate it. Doing so would significantly strengthen the paper. Another reason for not thinking this method is practical is that you are assuming access to w* in your experiments, which is clearly very unrealistic.
> > >
> > > The statement "Not going to work in any practical scenarios" is quite strong. In our paper, we have dedicated substantial effort to explaining why solely learning from humans can lead to dangerous outcomes. Building AI systems capable of inferring and correcting human false beliefs is a logical and necessary development.
> > >
> > > Regardless of whether the claim can be proven, it is important to emphasize that it is NOT a claim made by our paper. Therefore, we do not feel obligated to present results in a practical domain. We have explicitly stated in the paper:
> > > *"Our goal is not to introduce a high-fidelity benchmark or propose a state-of-the-art method for practical alignment—such objectives are beyond the scope of this paper. Instead, we aim to (1) demonstrate our theory while highlighting the importance of teaching and (2) present a prototypical benchmark that can inspire future work."*
> > > We believe that the evidence provided in the paper adequately supports these claims.
> > >
> > > Finally, assuming access to w*  in the experiments is not unrealistic. In many tasks, it is feasible to provide the AI with a high-fidelity simulation of the environment, such as a map or a 3D scan of a house, a clone of a website, or a sandbox OS environment.
> > > > In general, I am disappointed that the authors' have not engaged with my critique around presentation issues in the paper that make it hard to follow.
> > >
> > > We agreed with many of your suggestions and decided to respond to matters that we felt clarification was needed. We apologize if our response upset you.
> > >
> > > We fully agree with you on the importance of clear presentation. Our intention was to convey that certain papers naturally require more time to fully understand, especially when they introduce novel settings and concepts, as ours does. While we acknowledge that there is room for improvement in making the paper more accessible, we hope you can appreciate that papers addressing new and less familiar problems often take longer to digest than those proposing solutions to well-established issues.

---

> > > > ### Author Response · Authors · 2024-11-30
> > > > **Suggestions on presentation**
> > > >
> > > > Below we reply to your individual suggestions on the presentation.
> > > >
> > > > > The paper directly dives into describing what the ostensible alignment is without presenting the setup being assumed. Is it an MDP, POMDP, CIRL like Dec-POMDP setup? My sense is it is CIRL like Dec-POMDP setup but I am not super confident in that.
> > > >
> > > > It can be viewed as a special case of CIRL, which is a special case of Dec-POMDP. But CIRL and Dec-POMDP are more complicated than our framework, so we decided to directly describe the framework. We will make references to these frameworks in the next revision.
> > > >
> > > > > The paper is often introducing terminology without properly defining them. For example, second para of section 3.1 introduces the concept of plan without defining what the plan is.
> > > >
> > > > It is called a "solution plan" in the paper, denoting a solution to the problem defined by the human. We agree that "plan" can be difficult to understand in this context since there is no real environment introduced.
> > > >
> > > > > In description of ostensible alignment process, it is not clear to me how to think about 'conversation context'. Is it something like 'initial state' in MDP framework?
> > > >
> > > > Yes.
> > > >
> > > > > Bold p is used for joint policy, but for individual policies the p used is not bold. Authors should be consistent in their use of bold. I would also suggest avoiding the use of p and using some greek/latin symbol for policy as p is sort of a universal symbol for policy distribution. The convention in RL is to use \pi, but the authors are using it already.
> > > >
> > > > Joint policy is bold because it is a tuple (or vector) of two policies. The individual policies are not bold because they are not tuples. We use \pi for the plan because it is actually a policy in the real-world MDP.
> > > >
> > > > > Equation 1 was highly confusing for me, RHS is the definition of J^H(p) but it looks as the definition of max J^H(p).
> > > >
> > > > Thanks. We will fix it.
> > > >
> > > > > \mathcal{U^H}, \mathcal{U^A} are never defined.. presumably they are the sets from which utterances are sampled?
> > > >
> > > > We thought it was quite obvious that when writing x in X then X is the set of all possible xs. We will add clarifications for these terms.
> > > >
> > > > > I think if the authors lead the discussion with some sort of example that grounds these concepts, that would make it much easier for the reader to understand them.
> > > >
> > > > We have provide examples in the introduction. We plan to add more examples in the next revision.
> > > >
> > > > > Would not it make sense to start by introducing practical alignment; and then discussing 'ostensible' alignment as a simpler case?
> > > >
> > > > Ostensible alignment is in general NOT a simpler case of practical alignment, because practical alignment make specific assumptions about the cognitive model of the agents. We actually considered the option of introducing practical alignment first but the disadvantages are (1) as said, we cannot convey the generality of ostensible alignment's assumptions on the agents' cognition model (2) introducting the intricate cognition model of practical alignment may easily overwhelm readers. Ostensible alignment is simpler and easy to understand. So we choose to introduce it first and later extended it to practical alignment.

---

> > > > > ### Comment · Reviewer_8woS · 2024-12-01
> > > > >
> > > > > Your response did not upset me, and I did not think there was anything upsetting in your rebuttal.
> > > > >
> > > > > Regarding the point about 'practical method'; as noted in my original review, I don't see it as a weakness but at the same time, it does weaken the contribution. Authors' responded by asserting that they do provide a practical method, threby implicitly claiming it as a contribution *during* the rebuttal (I agree it is not claimed as a contribution in the original paper). Once a contirbution has been claimed, the onus is on authors' to justify its 'truthfulness'. Hence, I would stand by the claim in my original response that the onus is on authors to prove that their method is practical if they believe so.
> > > > >
> > > > > Relatedly, I am quite sympathetic to the comments of reviewer yQ that it is currently not obvious that this formalization is an obviously useful one. The 'intuitions' formalized in this work are common knowledge among alignment researchers, hence, the main value of this work would be in developing a formalism that either yeilds novel insights, or allows measurable progress on this very hard problem. In other words, the framework is only valueable to the extent it is the 'right' framework as yQ noted. I think developing a practical method based on this framework would be an obvious way to combat this concern.
> > > > >
> > > > > Thanks for replying to my specific complaints regarding the presentation; I would have strongly preferred if they were already incorporated in the paper though.
> > > > >
> > > > > In general, I believe my concerns regarding the presentation still stand, and I have developed novel concerns regarding the utility of this work to the community in the long run. Hence, even though I appreciate that this work exists and the authors are trying to tackle a very hard problem; I am going to keep my score.

---

> > > > > > ### Author Response · Authors · 2024-12-03
> > > > > > **Thank you for your response. Final post.**
> > > > > >
> > > > > > Thank you for your reply.
> > > > > >
> > > > > > Regarding the presentation concern, we apologize for not being aware that the manuscript could be updated during the rebuttal period. By the time our reply was posted, it was too late to make updates. Nevertheless, since you mostly agree with our reply to the presentation suggestions, we do not anticipate making major structural changes. Most improvements will focus on providing clearer definitions of the mathematical notations.
> > > > > >
> > > > > > Regarding the impact concern, we would like to reiterate our major contributions:
> > > > > >
> > > > > > 1. We construct a theoretical alignment problem in which learning-only approaches are *provably* insufficient.
> > > > > > 2. We demonstrate that AI systems must also be able to *teach* humans to solve this problem.
> > > > > > 3. We prove that learning-only approaches incentivize AI systems to *harmfully alter human beliefs about the world*, preventing them from understanding the truth about the world.
> > > > > > 4. We design a prototypical benchmark for teaching, demonstrate the effectiveness of teaching on this benchmark, and highlight the difficulty of the real-world problem by showing the imperfect performance of large language models (LLMs) on this benchmark.
> > > > > >
> > > > > > We believe that the theoretical results (1-3) are both novel and significant. They would provide rigrous formal arguments on the limitations and drawbacks of learning-only approaches like RLHF. Our arguments are highly general and applicable across various domains. This distinguishes them from intuitions gained through reasoning or empirical observations on a narrow set of domains, which may be widely known but have limited generalizability. Given the strong enthusiasm for RLHF and its variants, our negative theoretical results could have a profound impact by challenging current beliefs about the promise of these methods and pointing toward a new direction that dramatically differs from the current one.
> > > > > >
> > > > > > Specifically, the idea of building AI that can teach humans is relatively new to the alignment community. It was not mentioned in the popular survey on RLHF [1] (see list of proposed solutions in their section 4.2). Prior work that addresses the imperfection of human beliefs, such as Inverse Reward Design [2], relies on solutions based only on uncertainty. We observe that current alignment pipeline relies on manual effort to ensure that humans are aligned with reality. *What if we can build AI systems to automate this process for us?* (note that this is different from scalable oversight, which is about automating evaluation of AI systems). By illustrating this new idea, we aim to open up novel research challenges and opportunities, sparking new approaches to better address alignment.
> > > > > >
> > > > > > Last but not least, the benchmark we built, while being simple, demonstrate principles to overcome the challenges in simulating human cognition-based communication and exposing the weaknesses of LLMs. Moreover, it is actually harder than it appears to be, as LLM performance on the benchmark remains imperfect.
> > > > > >
> > > > > > [1] https://arxiv.org/abs/2307.15217
> > > > > >
> > > > > > [2] https://arxiv.org/abs/1711.02827

---

### Official Review · Reviewer_6rpm · 2024-11-04

**Soundness:** 3
**Presentation:** 2
**Contribution:** 3
**Rating:** 8
**Confidence:** 2

**Summary:**

This work presents a theoretical framework called practical alignment, which aims to address some of the limitations of the standard RLHF paradigm. Specifically, the standard RLHF framework assumes that human preferences accurately reflect human desires and that preferences remain constant over time. In contrast, this work presents a framework that does not assume that human preferences are static but are influenced by human beliefs about the world. This allows modeling scenarios where human beliefs are false or subject to change over time. By explicitly differentiating between human beliefs (mutable) and desires, this work introduces an optimization problem that naturally encourages an AI system to not only learn from a human, but also to correct their beliefs. Finally, this work presents an experimental framework called MindGrid to evaluate alignment scenarios. Here, results show that while LLMs can outperform naïve baselines, there is still significant scope for improvement.

**Strengths:**

1. **Clear writing**: The paper is well-written and generally easy to follow. The related work is thorough, and the overall flow is smooth.
2. **Novel Framework**: This paper presents a novel framework for alignment that aims to fix some of the well-known issues with the standard RLHF paradigms. Although there are limitations with the new framework, it presents an interesting lens on alignment. Notably, the work presents an example of how deceptive/manipulative AI agents can emerge from ostensible alignment, which is a key topic of interest in the community.

**Weaknesses:**

1. **Mathematical notation**: The mathematical notation at times can be more complicated than needed and I believe that mathematical clarity can be improved.

**Questions:**

1. In the practical alignment framework, it is assumed that the humans know $\psi^H$, the parameters for their desire function. If this is the case, then what prevents the AI agent from directly learning the desire function and optimizing that. In other words, what is the difference between the descriptive reward function of the ostensible framework and the desire reward function of the practical framework? I think giving more concrete examples here can be very helpful.
2. The presented framework assumes that human preferences might change with their belief about the world, but their desires stay constant. Although this is very abstract, it is possible for desires to change with beliefs about the world. I think it comes down to the modeling assumptions and there might not be one right answer, but I am curious to know the authors’ thoughts on this.
3. I see this framework as being more applicable to embodied AI systems that are interacting with the real world. These settings would naturally have many instances where a human’s internal world model is different from the real world. However, is this framework also applicable in the digital setting (such as AI chatbots)? What do the authors consider to be a world model here, what are concrete instances where the human’s model is incorrect?
4. The LLMs are specifically prompted that the human’s plan is outdated and that changes have been made to the grid. An ideal benchmark would be one where the LLM uses the human’s plan, infers that the world has changed and proceeds to correct the human’s belief about the world. What were the motivations behind the design decisions of this benchmark?
5. Is the communication model (discussion phase) described in this work new or has it been used in standard practice?
6. Why is it assumed that a human is also optimizing their communication policy? Does anything change if it is assumed that the human has a fixed policy? It might make notation simpler if it is not a necessary part of the framework.

---

> ### Author Response · Authors · 2024-11-22
> **Thank you for your feedback**
>
> Below are the answers to your questions:
>
> >1. In the practical alignment framework, it is assumed that the humans know ψ^H, the parameters for their desire function. If this is the case, then what prevents the AI agent from directly learning the desire function and optimizing that. In other words, what is the difference between the descriptive reward function of the ostensible framework and the desire reward function of the practical framework? I think giving more concrete examples here can be very helpful.
>
> The desire function is a parameter of the reward function. For example, as you evaluate my paper, your *reward function* determines the score you assign to it. But to decide that score, you could be simulating how impactful my paper is going to be in the future. The impact of my paper is measured by what we call a desire function. The word "desire" suggests that is a deeper, more latent motivation that influences your evaluation.
>
> The desire function alone does not fully define the reward function. The reward function is also dependent on a world model. If this world model is the real world, the reward function is normative. If the world model is imagined by the human, the reward function is descriptive.
>
> Even if the AI perfectly learns the desire function, it may still have imperfect knowledge of the reward function. Optimizing the right desire function in a wrong world model can still result in a suboptimal plan or policy. This is illustrated in Figure 1a: if the robot chooses Plan A, it optimizes the human's desire function in their (incorrect) model of the environment. That plan is suboptimal in the real environment.
>
> > 2. The presented framework assumes that human preferences might change with their belief about the world, but their desires stay constant. Although this is very abstract, it is possible for desires to change with beliefs about the world. I think it comes down to the modeling assumptions and there might not be one right answer, but I am curious to know the authors’ thoughts on this.
>
> This is a great question. This work presents a two-level hierarchy. You have preference over plans being influenced by beliefs and preference over world state-actions (the desire function). We assume a static desire function so that the objective is well-defined. A generalization of this framework would have more levels of abstraction. A desire can be driven by more abstract beliefs and desires. For example, you desire to be rich because you believe it would make you happy. Here, wanting to be happy is a more abstract desire than wanting to be famous. The most abstract desires, like wanting to be happy or to survive, may rarely change.
>
> > 3. I see this framework as being more applicable to embodied AI systems that are interacting with the real world. These settings would naturally have many instances where a human’s internal world model is different from the real world. However, is this framework also applicable in the digital setting (such as AI chatbots)? What do the authors consider to be a world model here, what are concrete instances where the human’s model is incorrect?
>
> The world model represents any assumptions that a human makes about the world. Humans make false assumptions all the time. For example, you chat with your virtual assistant to figure out the best gift for your loved one. You tell the assistant to buy an iPhone without knowing that they already ordered one yesterday. Rather than blindly executing the order, we want the assistant to correct your false beliefs by saying: “Your loved one has already purchased an iPhone”

---

> > ### Author Response · Authors · 2024-11-22
> > **Answers (continued)**
> >
> > > 4. The LLMs are specifically prompted that the human’s plan is outdated and that changes have been made to the grid. An ideal benchmark would be one where the LLM uses the human’s plan, infers that the world has changed and proceeds to correct the human’s belief about the world. What were the motivations behind the design decisions of this benchmark?
> >
> > The prompt is part of the solution, not the problem. The problem is exactly what you described: given a human plan and the current world, infer the false beliefs. Telling the model that the plan is outdated can be seen as a way of injecting human knowledge into the models. Even with that information, the models still struggle.
> >
> > > 5. Is the communication model (discussion phase) described in this work new or has it been used in standard practice?
> >
> > It depends on how you view it. If you view the discussion phase as an MDP, then it is not novel. But this MDP has special structures: the actions are joint utterances of two agents, and the reward function is constituted by parameters that are partially observed by one agent. To our knowledge, this specific discussion MDP has not been proposed anywhere.
> >
> > > 6. Why is it assumed that a human is also optimizing their communication policy? Does anything change if it is assumed that the human has a fixed policy? It might make notation simpler if it is not a necessary part of the framework.
> >
> > This is standard in cooperative decision-making and game theory. CIRL—a well-known alignment framework— formulates a similar objective which searches for the best human-robot policy pair. We are following their formulation. Assuming a static policy makes the formulation less general because in reality, humans do adapt their communication strategy based on the agent’s behavior.

---

> > > ### Comment · Reviewer_6rpm · 2024-11-30
> > >
> > > I thank the authors for their response. I think this paper correctly identifies flaws with existing preference formulations and proposes a new framework to address them. I do not think that their proposed framework is perfect, but there is no perfect framework for alignment and this paper offers a new lens on it. There also needs to be more empirical validation to understand the practical usefulness of framework, particularly in more complex real-world tasks, but I think that is outside the scope of this paper (for example, even CIRL had relatively simple experiments).
> > >
> > > Overall, I think this paper would be a valuable addition to the conference and might inspire some deeper discussion on alignment. I have raised my score accordingly.

---

> > > > ### Author Response · Authors · 2024-11-30
> > > > **Thank you for your consideration**
> > > >
> > > > We thank you for your appreciation of the paper. As you have said it really well, the paper is not perfect and we hope that it will provide a useful perspective. We wonder if your confidence has also changed. If so, would you consider reflect that in the confidence score so that your opinion will be weighted more strongly?

---

### Official Review · Reviewer_yQLP · 2024-11-04

**Soundness:** 3
**Presentation:** 2
**Contribution:** 1
**Rating:** 5
**Confidence:** 3

**Summary:**

The paper proposes a formalization of the AI alignment problem that distinguishes ostensible alignment (where human and AI agree on a target and the AI follows it) and practical alignment (where the target is additionally normatively correct). The paper proposes a few theoretical results in this framework as well as experiments in a Gridworld environment with simulated human language feedback to illustrate the proposed framework.

**Strengths:**

The paper is addressing an important problem: clarifying and formalizing the target of "AI alignment". The proposed distinction between ostensible alignment, inner alignment, and normative alignment is clear and adds conceptual clarity. The insight that humans can be mistaken when giving feedback to AI agents is importantly correct and the paper has some useful ideas for addressing this.

The proposed environment is interesting because it combines an agentic task with a language interface, which allows to test agentic alignment concepts with current language model agents.

**Weaknesses:**

I don't think the proposed framework is very insightful on it's own and the paper did not convince me that it is a useful formalization. The essential new component (that is not covered by previous formalizations like CIRL) seems to be that humans can be uncertain about their own preferences. This uncertainty can lead to the agent being "ostensibly aligned" but not practically aligned and this gap can get worse over time. While I agree that this is an important problem, the formalization does not seem to add anything that helps to solve this problem.

To be convinced that the formalization is useful, it would have to result in new insights that are not clear from the problem definition, or it would have to allow deriving quantitative bounds on the different misalignment gaps, and compare different methods. Both of these are missing from the current paper.

I also find the way the framework is set up not really mirrors how eg. RLHF is used in practice. The preference learning setting considered in the paper has a human pick between plans proposed by the agent (Figure 1). However, RLHF more commonly has humans pick between actual trajectories (eg., in Christiano 2017). If the human picked between two trajectories from the real world, the problems in eg. Fig 1 would not occur in this way. The paper discusses the human world model being wrong as a fundamental problem, but does not acknowledge that the extend to which this is problematic has a lot to do with the specific RLHF setup that is chosen.

While the experiment setup is interesting, the environments are very limited. There are two basic gridworld environments with a language interface. The experiments are not surprising and do not seem to be answering research questions posed by the practical alignment terminology. Rather, the experiments seem closer to the start of a benchmark. I think a benchmark built on these environments could be interesting, but the current setup is too limited.

So in summary, I don't think the contribution in this paper is large enough to warrant acceptance. I could reconsider my assessment if I am convinced the proposed framework is more useful, for example because I did not understand an important aspect of it.

**Questions:**

* Could you elaborate on why you find the proposed framework useful compared to prior frameworks for formalizing alignment?
  * What kind of results does your framework allow you to derive that prior work cannot study?
  * How could this work help to solve the proposed alignment problems?

---

> ### Author Response · Authors · 2024-11-22
> **Thank you for your review**
>
> We will first answer your two major questions:
>
> **1. How could this work help to solve the proposed alignment problems?**
>
> We want to emphasize that this is not a typical paper that proposes solutions to a well-established problem.  Instead, our goal is to introduce readers to a new problem and provide the **theoretical and empirical tools** needed to begin researching it. Specifically, we:
>
> - *Mathematically formulate the problem*. This is a prerequisite for developing theoretically grounded approaches. Without a mathematical formulation, the development of solutions is limited to relying on crude intuitions.
> - *Explain why this problem is significant* by formally proving approaches that do not recognize the problem leads to undesirable outcomes (section 5.2 & 5.3).
> - *Characterize the general properties of the solution*: it must involve teaching humans in addition to learning from them. This brings to attention a new set of problems orthogonal to learning from human feedback, opening new research opportunities.
> - *Build a toolkit to empirically study this problem (MindGrid)*. It might be deceptively simple but it is the first of its kind. Simulating communication with humans is extremely challenging. It is not just a matter of generating good-looking conversations; one needs to simulate the derivation of the communication intentions, which depends on various mental representations. On the other hand, if the simulation is too complex, the problem might be impossible to solve with the current methods. MindGrid achieves the balance: it is challenging enough for making research progress, but is simple enough to solve.
> - *Propose a solution to the problem*: we design AI that can teach humans. We use LLMs to enable such capability. Our experiment results clearly suggest that AI that can also teach humans outperform any AI that only learns from them.
>
> **2. What kind of results does your framework allow you to derive that prior work cannot study?**
>
> Below we list our results that are unattainable by previous frameworks:
> - *Alignment requires AI that can teach humans*. Previous frameworks implicitly assume humans have perfect world models, so teaching is unnecessary. We believe that this is an important result, as it points to a set of problems that are currently overlooked.
> - *The relationship between the human’s knowledge gap and the suboptimality of the performance of the AI*. Concretely, the AI performance suboptimality gap scales with a rate between Omega(H*D) and O((H^2)*D) where H = 1/(1 - gamma) is the effective horizon and D is the divergence between the human’s world model and the real world (theorems 4.1 and 5.2). Previous frameworks do not model the human’s world model, so such a result was not possible.
> - *AI systems trained with previous frameworks are incentivized to manipulate humans* (theorem 5.3).  This problem does not exist if humans have perfect world models, which is assumed in previous frameworks. If a human knows everything about the world, the optimal behavior for an AI is to respect their beliefs and solely focus on learning.
>
> We note that while some of these results may not be surprising to you, writing them down in precise mathematical terms takes effort and was not done before to our best knowledge.

---

> > ### Author Response · Authors · 2024-11-22
> > **Clarifications**
> >
> > We now clarify some of the misunderstandings about the paper
> >
> > > The essential new component (that is not covered by previous formalizations like CIRL) seems to be that humans can be uncertain about their own preferences. This uncertainty can lead to the agent being "ostensibly aligned" but not practically aligned and this gap can get worse over time.
> >
> > Our paper is not about humans being uncertain about the world. It is about humans having **incorrect** models of the world. They are *confidently* mistaken.
> >
> > > While I agree that this is an important problem, the formalization does not seem to add anything that helps to solve this problem.
> >
> > Please see our response in the previous post. We believe that the first step towards rigorous AI science is to be able to mathematically describe a problem. Such a formulation allows making **formal claims** instead of intuition-based claims. Since the studied problem is naturally complex, formulating it is not an easy task.
> >
> > Before talking about solutions, it is important to  convince readers that this problem cannot be solved by existing methods. We do that by (1) proving that RLHF approaches can lead to dangerous outcomes and (2) demonstrating that LLM teaching capabilities are imperfect. In particular, (1) is possible thanks to our mathematical formulation of the problem. These results help motivate and guide the search for better solutions.
> >
> > Last but not least, toward the end of the paper, we contribute a solution idea and prototype it in a simple environment.
> >
> > > To be convinced that the formalization is useful, it would have to result in new insights that are not clear from the problem definition, or it would have to allow deriving quantitative bounds on the different misalignment gaps, and compare different methods.
> >
> > The main insight is clearly articulated in the paper's title: learning alone is insufficient to achieve alignment. In contrast, existing formulations assume that learning is enough, which can lead to the development of dangerous AI systems. This realization cannot be formally proved without our problem formulation. Additionally, we provide bounds on the misalignment gap—something that cannot be derived using existing formulations. We would greatly value understanding why these contributions are not recognized. It would be helpful if you could provide examples of what you were expecting or looking for.
> >
> > > I also find the way the framework is set up not really mirrors how eg. RLHF is used in practice. The preference learning setting considered in the paper has a human pick between plans proposed by the agent (Figure 1). However, RLHF more commonly has humans pick between actual trajectories (eg., in Christiano 2017).
> >
> > Our framework generalizes RLHF by enabling the human and the AI system to exchange **any type of feedback**, including trajectory evaluations. Specifically, during the discussion phase, both parties can send and receive any form of information through utterances (u_t). The R(plan) we define represents **the objective of the problem**, NOT the actual feedback exchanged between the two parties. In Christiano et al. (2017), while the feedback consists of trajectory preferences, the goal is the same as ours: find a plan (a policy) that maximizes some R(plan)—the performance of the policy in an environment.
> >
> > Moreover, in Figure 1, the environment is deterministic, so a plan of actions completely determines a trajectory, and either of these fully defines a deterministic policy. We will clarify this point in the paper.
> >
> > > If the human picked between two trajectories from the real world, the problems in eg. Fig 1 would not occur in this way.
> >
> > The problem persists even if trajectories are drawn from the real world because **the human does not fully observe the world**. Consider this example: you ask a robot to buy groceries at a supermarket, expecting the bill to be 100 USD. The robot returns home with a bill of 150 USD and provides a video documenting its shopping process. Although the video was honestly taken in the real world, you did not witness its creation, so there is a chance you might not believe it. You might suspect that the robot fabricated the video to steal 50 USD from you. Similar to our example, the robot is faced with a dilemma: respecting the human's world model makes the task impossible, but disrespecting it would upset the human (without proper teaching).

---

> > > ### Comment · Reviewer_yQLP · 2024-11-25
> > >
> > > Thank you for the clarifications and responding to my questions.
> > >
> > > I certainly think it is valuable to identify and formalize open problems and I do not think a good paper needs to only solve problems. My main concern is more that it is easy to come up with _some_ formalization of a problem, but one needs to think about what is the _right_ formalization of a problem in order to make progress.
> > >
> > > The paper correctly identifies there is a problem in learning from human feedback that human's can be mistaken about the world. However, this is well known, and not a novel contribution (eg. see [1] and citations therein). The paper then goes on to provide a straightforward formalization by distinguishing the "real" transition model from the human's model of the transitions -- this is novel for this problem, but very analogous to what's typically done when formalizing model-based RL.
> > >
> > > However, the paper, in my opinion, fails to convincingly show that this formalization is useful or insightful. None of the results are surprising or provides a new angle for solving the proposed problem -- or at least the discussion doesn't make the new insights clear enough.
> > >
> > > While overall I remain skeptical, your response helped clarify some of the confusions I had about the paper and I am now more optimistic about the proposed formalization. I will increase my score from 3 to 5, but I still think the present paper is below the bar for an ICLR publication. At the very least, I think the paper would need to be rewritten to be clearer about the usefulness of the proposed formalism.
> > >
> > > [1] Casper, Stephen, et al. "Open problems and fundamental limitations of reinforcement learning from human feedback." arXiv preprint arXiv:2307.15217 (2023).

---

> ### Author Response · Authors · 2024-11-30
> **Thank you for your feedback**
>
> We appreciate the time and effort you spent on evaluating our paper. We would like to reply to some of your points. Please understand that we are not pressuring to change your evaluation, but we think these clarifications might be helpful for the AC to make the final decision.
>
> > The paper correctly identifies there is a problem in learning from human feedback that human's can be mistaken about the world. However, this is well known, and not a novel contribution (eg. see [1] and citations therein).
>
> We believe that the familiarity of a natural phenomenon does not diminish the value of a mathematical framework proposed to model it. For example, while everyone knows the sun rises in the east, writing an equation to explain and predict the sun's morning movement would be considered a marvelous scientific achivement. We think that it is a bit unfair to compare our mathematical framework with [1], a paper that only describes natural phenomena but provides no mathematical formulations. It would be more convincing if you presented a paper that proposed a mathematical formulation of the problem we study. There are indeed a few and we have compared with them in the related work section.
>
> Of couse, novelty is not enough. As you said, we also need to convince readers why our formulation is useful. We reply to this point below.
>
> > [the formulation is] analogous to what's typically done when formalizing model-based RL.
>
> We think that this should be regarded as a strength rather a weakness of the proposed formulation. The formulation enables us to borrow theoretical tools from model-based RL to derive theoretical insights. The fact that the results appear intuitive and natural  is evidence that the problem is formulated in the right way.
>
> Note that our problem is different from model-based RL. Model-based RL concerns how the **error of an AI's world model** affects its performance. There is no interaction with humans. We extend this framework to relate the **error of the human's world model** with the performance of an AI that learns from them.
>
> > However, the paper, in my opinion, fails to convincingly show that this formalization is useful or insightful. None of the results are surprising or provides a new angle for solving the proposed problem -- or at least the discussion doesn't make the new insights clear enough.
>
> In addition to connecting with model-based RL, our formulation leads to these insights:
> * Alignment requires AI capable of teaching humans. This idea of building AI to influence humans initially seems taboo, as current approaches treat the human reward function as “sacred”, untouchable information. Our formulation recognizes that humans have multiple layers of will: while deeper desires should be respected, preferences shaped by beliefs can—and must—be corrected to avoid undesirable consequences. If you look at the list of solutions proposed by [1] (section 4.2), you will not find any similar ideas. Our solution idea goes beyond conventional thinking.
> * A simple but important result (theorem 5.3), which states that AI can be unintentionally trained to manipulate human's beliefs about the world. Recently several papers have provide empirical evidence of this phenomenon (e.g., https://arxiv.org/abs/2411.02306, https://arxiv.org/abs/2409.12822). Our result formally explains (or predicts) these empirical phenomenon. The result is possible because we incorporate the human's world model in our formulation.

---

### Author Response · Authors · 2024-12-03
**General response**

We thank all the reviewers for the valuable feedback and conversations.

Among four reviewers, two are highly positive about the value of the paper. The other two raise two major concerns: (1) the presentation of the paper (2) the usefulness and impact of the proposed formalization.

Regarding the *presentation concern*, we apologize for not being aware that the manuscript could be updated during the rebuttal period. By the time we realized that, it was too late to make updates. Nevertheless, we do not anticipate making major structural changes. Most improvements will focus on providing clearer definitions of the mathematical notations, as well as the contributions and implications of the paper.

Regarding the *impact concern*, we would like to reiterate our major contributions:

1. We construct a theoretical alignment problem in which learning-only approaches are provably insufficient.
2. We demonstrate that AI systems must also be able to teach humans to solve this problem.
3. We prove that learning-only approaches incentivize AI systems to harmfully alter human beliefs about the world, preventing them from 4. understanding the truth about the world.
4. We design a prototypical benchmark for teaching, demonstrate the effectiveness of teaching on this benchmark, and highlight the difficulty of the real-world problem by showing the imperfect performance of large language models (LLMs) on this benchmark.

We believe that the theoretical results (1-3) are both novel and significant. They would provide rigrous formal arguments on the limitations and drawbacks of learning-only approaches like RLHF. Our arguments are highly general and applicable across various domains. This distinguishes them from intuitions gained through reasoning or empirical observations on a narrow set of domains, which may be widely known but have limited generalizability. Given the strong enthusiasm for RLHF and its variants, our negative theoretical results could have a profound impact by challenging current beliefs about the promise of these methods and pointing toward a new direction that dramatically differs from the current one.

Specifically, the idea of building AI that can teach humans is relatively new to the alignment community. It was not mentioned in the popular survey on RLHF [1] (see list of proposed solutions in their section 4.2). Prior work that addresses the imperfection of human beliefs, such as Inverse Reward Design [2], relies on solutions based only on uncertainty. We observe that current alignment pipeline relies on manual effort to ensure that humans are aligned with reality. What if we can build AI systems to automate this process for us? (note that this is different from scalable oversight, which is about automating evaluation of AI systems). By illustrating this new idea, we aim to open up novel research challenges and opportunities, sparking new approaches to better address alignment.

Last but not least, the benchmark we built, while being simple, demonstrate principles to overcome the challenges in simulating human cognition-based communication and exposing the weaknesses of LLMs. Moreover, it is actually harder than it appears to be, as LLM performance on the benchmark remains imperfect.

[1] https://arxiv.org/abs/2307.15217

[2] https://arxiv.org/abs/1711.02827

---

### Meta-Review · Area_Chair_RnEH · 2024-12-22

**Metareview:**

The paper introduces a new mathematical framework called "practical alignment" that formalizes several weaknesses of RLHF and proposes theoretical ways of addressing them. It also introduces a new toolkit for evaluating alignment scenarios. It also presents a study with a simulated human agent and an AI agent to illustrate the benefits of AI teaching humans during the alignment process.

Strengths:

- The importance of the problem the paper is trying to address
- A novel mathematical framework for describing alignment, which predicts weaknesses of RLHF
- A new toolkit, MindGrid, for analyzing alignment scenarios
- The idea that AI teaching humans is needed for practical alignment

Weakness:

- Lack of empirical evidence in favor of the practical alignment framework, including evidence that practical alignment actually results in better alignment than existing alignment methods
- Lack of an algorithm for practical alignment
- Experiments only in simulated settings

While the paper introduces some interesting ideas, the metareviewer thinks it is not ready for publication, for reasons somewhat similar to reviewer *yQLP*'s. Namely, all strict mathematical models of human behavior make abysmally inaccurate assumptions. This goes for the practical alignment framework as well:

- It assumes that alignment entails learning a reward function, whereas it doesn't. E.g., DPO doesn't learn one. This leads the rest of the paper to focus on rewards; the benefits of this are unclear.

- In the practical alignment formalization (section 3.2), the paper assumes the AEAP cognition model. It's not clear why this model is a good choice.

- In general, in its practical alignment formalization, the paper assumes that people have use linear additive utility. It's a common and mathematically convenient but wildly unrealistic assumption.

None of this means that practical alignment is a bad framework, but it does make it non-obvious that, when all its unrealistic assumptions are combined, practical alignment will work better than the ostensible one. The only way to find out is to run an extensive study, involving real humans, using an algorithm that optimizes practical alignment, but the paper doesn't propose such an algorithm. Moreover, the presented study is very artificial and doesn't involve real humans.

Note that this algorithm per se isn't a precondition for publication, but it seems indispensable for assessing the merits of the proposed framework, which *is* crucial for publication. As two of the reviewers have mentioned, currently the framework makes predictions that either well-known drawbacks of RLHF (contrary to the authors' claim in the rebuttal, one doesn't need mathematical formalisms to identify these drawbacks) or that the paper verifies in purely synthetic experiments (the benefits of AI teaching people during alignment). Moreover, it ignores the very common case of alignment failure where the human judge lacks the information to form an accurate preference and the AI doesn't have that information either, so it can't teach the human about it. All this makes the proposed framework interesting but too lacking in empirical support to be accepted at ICLR.

**Additional Comments On Reviewer Discussion:**

The reviewers unanimously agreed on the importance of the alignment problem the paper's framework tries to address but differed on its value for actually addressing that problem, on the value of the MindGrid toolkit, and even on the paper's clarity. The discussion helped address some of these issues, but the doubts about the impact potential of the proposed framework have remained, and the metareviewer finds them sufficiently serious to make publication premature.

---

### Decision · Program_Chairs · 2025-01-22

Reject